# Molecular and clinical profiling in a large cohort of Asian Indians with glycogen storage disorders

**Tejashwini Vittal Kumar**[1], **Meenakshi Bhat**[2,3], **Sanjeeva Ghanti Narayanachar**[3], **Vinu Narayan**[2], **Ambika K. Srikanth**[1], **Swathi Anikar**[1], **Swathi Shetty**[1] *

**1** Molecular Genetics, Centre for Human Genetics, Bengaluru, India, **2** Clinical Genetics, Centre for Human Genetics, Bengaluru, India, **3** Pediatric Genetics, Indira Gandhi Institute of Child Health, Bengaluru, India

* swathi@chg.res.in

**Data Availability Statement:** All relevant data are within the paper and its Supporting information files.

## Abstract

Glycogen storage disorders occur due to enzyme deficiencies in the glycogenolysis and gluconeogenesis pathway, encoded by 26 genes. GSD's present with overlapping phenotypes with variable severity. In this series, 57 individuals were molecularly confirmed for 7 GSD subtypes and their demographic data, clinical profiles and genotype-phenotype co-relations are studied. Genomic DNA from venous blood samples was isolated from clinically affected individuals. Targeted gene panel sequencing covering 23 genes and Sanger sequencing were employed. Various bioinformatic tools were used to predict pathogenicity for new variations. Close parental consanguinity was seen in 76%. Forty-nine pathogenic variations were detected of which 27 were novel. Variations were spread across GSDIa, Ib, III, VI, IXa, b and c. The largest subgroup was GSDIII in 28 individuals with 24 variations (12 novel) in *AGL*. The 1620+1G>C intronic variation was observed in 5 with GSDVI (*PYGL*). A total of eleven GSDIX are described with the first Indian report of type IXb. This is the largest study of GSDs from India. High levels of consanguinity in the local population and employment of targeted sequencing panels accounted for the range of GSDs reported here.

## Introduction

Glycogen Storage Disorders (GSDs) are caused by deficiencies in the enzymes involved in the glycogenolysis and gluconeogenesis pathway. The incidence of GSDs is 1 in 20,000–43,000 live births [1]. Based on the protein that is affected, GSDs are classified into 13 different types involving 26 genes [2]. This study describes 57 individuals of Indian origin with GSD type I, III, VI, IXa, IXb and IXc. GSDI is further divided into Ia, Ib, Ic and Id with GSDIa accounting for about 80% of all GSDI cases and resulting from pathogenic variants in the *G6PC1* gene [3]. The remaining 20% includes Ib, Ic and Id, all of which harbour pathogenic variations in the *SLC37A4* gene [3, 4].

GSDIa (OMIM 232200) is caused by deficiency of G6Pase-α which inhibits conversion of G6P to free glucose and phosphate [4]. This enzyme is expressed in the liver, kidney, and intestinal mucosal cells. GSDIa manifests early in infancy with severe hypoglycemia,

**Funding:** The author(s) received no specific funding for this work.

**Competing interests:** The authors have declared that no competing interests exist.

hyperlipidemia, hepatomegaly, elevated lactate, uric acid levels and most affected children develop a cherubic face [3].

GSDIb (OMIM 232220) is caused due to deficiencies in transport of glucose-6-phosphate (G6P) from the cytosol into the microsomal lumen by glucose-6-phosphate translocase (G6PT) encoded by the *G6PT/SLC37A4* gene [5, 6]. Clinical features of GSDIb are identical to GSDIa and include neutropenia and recurrent infections [5].

GSDIII (OMIM 232400) results from the defective functioning, or the absence of the glycogen debranching enzyme (GDE) encoded by the *AGL* gene whose deficiency causes storage of an intermediate form of glycogen called Limit Dextrin (LD) [7, 8]. GSDIII has an overall incidence of 1 in 100,000 [9, 10] but in some ethnic groups like the Faroe Island population and in Tunisians, the prevalence is as high as 1 in 3,600 and has been attributed to a founder mutation [11]. Similarly, Italian, and Mediterranean studies have reported a common splice site variation in nearly 28% of GSDIII individuals [10].

GSDIII is categorised into two subtypes, GSDIIIa and GSDIIIb and genotype–phenotype variability could be due to differential expression of GDE in different tissues [8, 9]. GSDIIIa, which accounts for up to 85% of GSDIII cases have muscle and liver involvement and most also have cardiac involvement of variable severity. The remaining 15% are classified as type GSDIIIb in whom only liver GDE is deficient [9, 10].

GSDVI (OMIM 232700) is caused by deficiency of the hepatic glycogen phosphorylase enzyme which has 3 different isoforms encoded by 3 different genes (*PYGL, PYGM* and *PYGB*). *PYGL, PYGB* and *PYGM* are expressed in the liver, brain, and muscle respectively [12], with *PYGL* variations being most frequently observed.

GSDIX is caused due to deficiency of the enzyme phosphorylase b kinase (PhK). PhK contains 4 subunits alpha (*PHKA1, PHKA2*), beta (*PHKB*), gamma (*PHKG1* and *PHKG2*), and delta (*CALM1, CALM2* and *CALM3*) [13]. *PHKA1* (GSDIXd–OMIM 300559) is expressed in the liver, *PHKA2* (GSDIXa- OMIM 306000) in muscle, and both are X-linked. *PHKB* (GSDIXb- OMIM 261750) is expressed in both liver and muscle, while *PHKG2* (GSDIXc-OMIM 613027) is only expressed in the liver and both these are inherited as autosomal recessive disorders [14].

GSDs have a considerable amount of heterogeneity in their clinical presentations and diagnosis has been largely dependent on invasive biopsies, biochemical assays and measurements of enzyme activity which is not routinely available. This study aims at diagnosing GSDs molecularly into different subtypes using NGS and Sanger technology keeping in mind the number of genes involved and to overcome the difficulties stated above. The study also intended to examine the spectrum of pathogenic variations in the various genes involved in order to add to the available molecular mutation database including the presence of any founder mutations or hotspots.

There are only four other Indian studies published thus far with the largest one including 24 individuals [4, 9]. None of these have implicated any hotspots or founder effect mutations.

## Material and methods

### Ethical clearance and consent to participate

This study was approved by the institutional ethics committee at the Centre for Human Genetics, Bangalore, India (CHG/077/2020-21/009). Informed consent from parents/ legal guardians of affected children was obtained.

### Clinical data

**Study design and setting.** The clinical and biochemical data for 57 individuals from one southern Indian centre were recruited over 15-year period.

**Clinical and biochemical screening.** The following clinical and biochemical criterions were recorded for each- age of onset, history of symptoms, height, weight, biochemical parameters including fasting blood glucose, creatine kinase, serum lipids, serum lactic acid, liver function profile, kidney function profile and liver biopsy reports where available. Presenting features were also recorded and included hepatomegaly with or without splenomegaly, hypoglycaemia with a history of seizures, diarrhoea, growth retardation and recurrent infections.

*Genetic analysis.* Genomic DNA was extracted from peripheral blood samples. For prenatal diagnosis, chorionic villus sampling or amniotic fluid was used. The coding region and the intron/exon junctions of *G6PC1*, *SLC37A4*, and *AGL* genes were amplified by PCR. Information for primer sequences for GSDIa, GSDIb and GSDIII were obtained from earlier studies [10, 15, 16] [Tables 1–3 in S1 File]. The amplified product was purified and sequenced using the standard protocol for ABI 3500 Genetic Analyser (Applied Biosystems). The results were analysed using Sequencher® software. A targeted NGS panel comprising of 23 genes (*AGL, ALDO, ENO3, G6PC1, GAA, GBE1, GYG1, GYS1, GYS2, LAMP2, LDHA1, PFKM, PGAM2, PGM1, PHKA1, PHKA2, PHKB, PHKG2, PRKAG2, PYGL, PYGM, SLC2A2 AND SLC37A4)* was used for testing other GSDs. Genes *CALM1*, *CALM2* and *PHKG1* were not included in this panel as no pathogenic variants have been identified in these genes thus far. The prepared DNA libraries were sequenced to a mean >80-100X coverage on the Illumina sequencing platform. All likely disease-causing variants were confirmed by Sanger sequencing.

## Bioinformatic analyses

The pathogenicity of novel mutations was assessed using online bioinformatic tools including Mutation Taster (MT2), Polyphen-2 (version—2.2.2), SIFT (version—5.2.2), Provean (version -1.1), Variant Effect Predictor (Ensembl release 104) for coding regions, and Human splice finder-HSF (http://www.umd.be/HSF3/) and Cryp-Skip (https://cryp-skip.img.cas.cz/) for splice site variations. The identified variants were checked in the 1000 Genome and dbSNP databases for allele frequencies. In addition, 100 control samples from normal individuals (ethnically matched unaffected adults) were screened for the novel variations reported in this study. Multiple sequence alignments were carried out using CLUSTALW OMEGA (http://www.genome.jp/tools-bin/clustalw) to analyse conserved regions across organisms sharing more than 75% sequence homology in all the genes studied.

Multiple programs like HMMTOP, MEMSAT, PHDhtm, TMHMM, TMpred, Topcons and TopPred were used to predict the membrane topology of *G6PC1* and *SLC37A4* from their respective amino acid sequences. The consensus results from the above were used for graphic illustration using the tool Protter (http://wlab.ethz.ch/protter/) (Fig 1).

## Results

A total of 57 samples consisting of 33 males and 24 females were molecularly confirmed as GSD's over a 15-year period. Sanger sequencing identified 21 variations and 28 variations were identified using a targeted NGS based GSD panel. Twenty-seven of the 49 variations detected were novel (Table 1). These included missense, nonsense, deletions, duplications, insertions, and splice site changes in the genes *G6PC1* (6), *SLC37A4* (5), *AGL* (24), *PYGL* (5), *PHKA2* (2), *PHKB* (3) and *PHKG2* (4) (Table 8 in S1 File). All observed novel variations were in regions that are conserved across species (Figs 1–6 in S1 File). Parental consanguinity was reported in 45 families (76%) in which 8 families had another affected sibling. The age at diagnosis in all sub-types ranged from 1m to 14y but in 6 individuals with GSDVI who presented with milder symptoms, the age at diagnosis ranged from 4-14y. Parents of 47 probands had confirmation of carrier status by Sanger sequencing. Prenatal counselling was done in eight

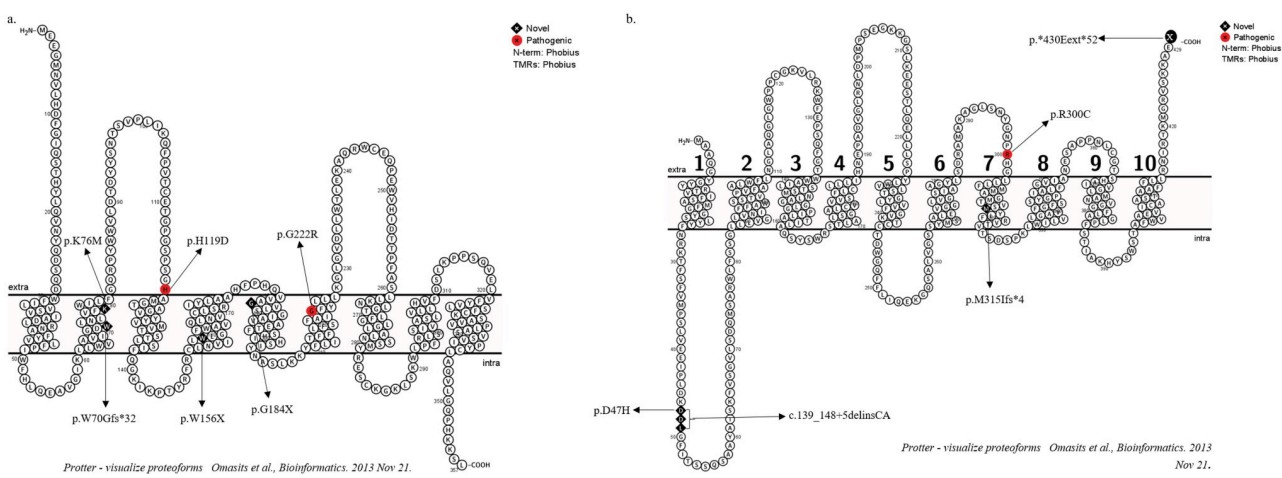

**Fig 1. Topology of (a) G6Pase-α enzyme and (b) G6PT enzyme showing all variations identified in this study.**

**Table 1. Novel variation identified in our study.**

| Patient | Age / sex | Consanguinity | GSD type | HGVS Nomenclature |
|---|---|---|---|---|
| P 03 | 5m/M | YES | Ia | NM_000151.4(G6PC1):c.227A>T (p.Lys76Met) |
| P 04 | 1m/M | YES | Ia | NM_000151.4(G6PC1):c.468G>A (p.Trp156Ter) |
| P 05 | 1 y/ F | YES | Ia | NM_000151.4(G6PC1):c.550G>T (p.Gly184Ter) |
| P 06 | 10m/F | Na | Ia | NM_000151.4(G6PC1):c.208del (p.Trp70fs) |
| P 07 | 1y/M | NO | Ib | NM_001164277.2(SLC37A4):c.945_964del (p.Met315fs) |
| P 08 | 14y/M | YES | Ib | NM_001164277.2(SLC37A4):c.1287_1290del (p.Ter430GluextTer?) |
| P 09 | 4m/M | NO | Ib | NM_001164277.2(SLC37A4):c.139G>C (p.Asp47His) |
| P 13 | 9y/ M | YES | III | NM_000642.3(AGL):c.1788T>G (p.Tyr596Ter) |
| P 14 | 3 y/ F | YES | III | NM_000642.3(AGL):c.2497C>T (p.Gln833Ter) |
| P 16 | | YES | III | NM_000642.3(AGL):c.3214G>T (p.Glu1072Ter) |
| P 18 | 1.6y/M | YES | III | NM_000642.3(AGL):c.4371T>G (p.Tyr1457Ter) |
| p 20 | 3.5y/M | NO | III | NM_000642.3(AGL):c.947_948del (p.Leu316fs) |
| P 21 | 7m/F | NO | III | NM_000642.3(AGL):c.2996del (p.Pro999fs) |
| P 22 | 2y/M | YES | III | Exon 30-31del |
| P 25 | 8 y/ F | YES | III | NM_000642.3(AGL):c.664+1G>C |
| P 28 | 2 y/ M | YES | III | NM_000642.3(AGL):c.1612-1G>A |
| P 31 | 3 y/ M | YES | III | NM_000642.3(AGL):c.2949+5G>A |
| P 32 | 6 y/ F | YES | III | NM_000642.3(AGL):c.2949+5G>A |
| P 37 | 6y/M | YES | III | NM_000642.3(AGL):c.3362G>A (p.Arg1121Lys) |
| P 38 | 1.2 y/M | YES | III | NM_000642.3(AGL):c.4599G>C(p.Ter1533Lysext*) |
| P 41 | 5y/F | NO | VI | NM_002863.5(PYGL):c.72C>A (p.Asn24Lys) |
| P 44 | 12y/F | YES | VI | NM_002863.5(PYGL):c.33dup (p.Arg12fs) |
| P 46 | 4y/M | NO | VI | NM_002863.5(PYGL):c.2056G>C (p.Gly686Arg) |
| P 49 | 21y/M | YES | IXb | NM_000293.3:c.(?_-1)_(1068+1_1069-1)del |
| P 50 | 2Y/M | YES | IXb | NM_000293.3:c.(76+1_77-1)_(1068+1_1069-1)del |
| P 51 | 6y/F | YES | IXb | NM_000293.3(PHKB):c.1364-2A>G |
| P 55 | 1.5Y/F | YES | IXc | NM_000294.3(PHKG2):c.229G>A (p.Glu77Lys) |

families seeking prenatal diagnosis [GSDIa (2+ 2 subsequent pregnancies in a single couple), GSDIb (1), GSDIII (3), GSD IXb (1)]. All three couples with previous GSDIII opted against prenatal testing and two of these were confirmed to be affected after birth. One with GSDIa and GSDIXb also declined prenatal diagnosis and went on to have an unaffected child each. Invasive testing with chorion villus sampling was done in the remaining three cases (GSDIa and b). All were unaffected and reviewed postnatally.

## Clinical description

Progressive abdominal distension caused by liver enlargement was the most common presenting symptom in 71% followed by hypoglycemic seizures in 15% (Table 2). Interestingly, incidental identification of abnormal liver function during investigation of an intercurrent illness accounted for GSD diagnosis in 12%. One case of Type IXb was diagnosed at 1m, because of a previously affected sibling. Thirty-seven children have been on dietary therapy for durations ranging from 6m to 8y. The majority of these were affected with GSDIII. Two individuals each with Type GSDIa and Type GSDIb are deceased. The cause of death was liver cell decompensation and severe infection, respectively.

## Glycogen storage disease type Ia

We identified 6 disease causing variations in *G6PC1*, of which 4 are novel. P01, an affected female presented with severe hypoglycemia, grossly elevated serum triglycerides and cholesterol with liver pathology showing distended hepatocytes containing PAS positive, diastase negative material. She had a homozygous missense variation in exon 3, c.355C>G [p. His119Asp], where histidine a basic residue located in the active site of *G6PC1* is replaced by acidic aspartic acid. P02 had a homozygous missense variation c.664G>A [p.Gly222Arg], reported as pathogenic and described in the NCBI database as 'likely pathogenic' [17]. The arginine at this position was shown to reduce enzyme production. P03 presented with

**Table 2. Demographic details of 57 affected individuals with GSD.**

| Type of GSD Total No. 53 | Gene mutated | Number of patients | M/ F | Parental Consanguinity, No. of families | Age range at diagnosis | First clinical symptom/ presentation | Other sibling affected | Dietary therapy | Outcome A: alive D: death NK: not known |
|---|---|---|---|---|---|---|---|---|---|
| 1a | *G6PC* | 6 | 3 M 3 F | 5 1 NK | 1m-3y | Abdominal distension 5 | 3/5 | 4 | 2 D 2 A 1 NK |
| 1b | *G6PT/ SLC37A4* | 4 | 4 M | 2 | 4m-14y | Abdominal distension 3 Recurrent infection 2 | 2/4 | No | 2 D 3 NK |
| III | *AGL* | 28 | 14 M 14 F | 21 | 7m-9y | Abdominal distension 20 Hypoglycemic seizures 8 | 3/28 | 21 | 22 A 6 NK |
| VI | *PYGL* | 8 | 4 M 4 F | 5 | 4y-14y | Abdominal distension 8 | 1/8 | 8 | 8 A |
| IXa | *PHKA2* | 2 | 2 M | 0 | 8m-2y | Abnormal lipid profile Abdominal distension Failure to thrive | No | 1 | 1 A 1 NK |
| IXb | *PHKB* | 3 | 2 M 1 F | 3 | 4y | Abdominal distension Affected sibling | 1 | No | 1 A |
| IXc | *PHKG2* | 6 | 3 M 3 F | 6 | 1 ½y-21y | Abdominal distension 7 Abnormal liver function 6 | No | 3 | 3 A 5 NK |

hepatosplenomegaly, raised liver enzymes, hyper triglycerides and penoscrotal hypospadias (having a normal karyotype). The individual had a novel c.227A>T [p.Lys76Met] variation in exon 1. P04 and P05 harboured novel homozygous nonsense variations in exon 4, c.468G>A [p.Trp156X] and c.550G>T [p.Gly184X]. They presented with biochemical abnormalities, grossly abnormal liver function, distended hepatocytes with vacuolated cytoplasm (PAS positive) and early cirrhosis on liver biopsy. Both died shortly after diagnosis, one aged 14m and the other 2y. Severe biochemical derangements and hepatic dysfunction were the reported cause of death in both. P06 had a homozygous novel deletion at position c.208delT in exon 2 leading to premature termination of protein synthesis after 32 amino acids due to a frameshift.

## Glycogen storage disease type Ib

Five different variations were identified in *SLC37A4* in 4 individuals (P07-P09). P07 harboured novel compound heterozygous variations c.139_148+5delinsCA in exon 1 and c.945_964del in exon 6 leading to premature termination of the protein. This patient had extensive skin pustulosis, tachypnoea, recurrent hypoglycemica and neutropenia. P08 had a novel deletion of c.1287_1290del [p.*430Gluext*52] which causes the reading frame to change, loss of translation at the 3' end, degradation of the mRNA and instability of the protein [18, 19]. Both had typical clinical features in keeping with the diagnosis of GSDIb. P07 died aged 22m and P08 at the age of 4y of sepsis.

P09 had a homozygous novel missense variation, c.139G>C [p.Asp47His] in exon 2. P10 had a previously reported pathogenic missense variation c.898C>T [p.Arg300Cys] in the major facilitator superfamily domain of the *SLC37A4* protein [20].

## Glycogen storage disorder III

GDE consists of 1532 amino acids (NP_000019.2) containing the transferase catalytic, the glucosidase catalytic residue, and the glycogen binding domains. Variations in these regions are known to affect protein functioning [10, 21]. In total, 24 variations were found in the *AGL* gene of which 12 are novel (Fig 2). 8 were in the above-mentioned domains. Affected individuals (14 males and 14 females) were from Karnataka (23), Andhra Pradesh (2) and West Bengal (3). The age at diagnosis ranged from 7m to 9y (median 2.7y, mean 3.7y). 64% had close parental consanguinity ranging from uncle-niece to second cousin marriages. Five had a similarly affected sibling and only one was of non-consanguineous parentage. Two of these had previously deceased siblings and the other three were identified as a result of family screening. Liver enlargement was documented in all and 21 (84%) had a liver biopsy at diagnosis. In the majority, liver biopsy showed enlarged hepatocytes with pale cytoplasm, centrally located nucleus, Periodic acid-Schiff (PAS) positive and diastase sensitive staining material and peri-portal inflammation with liver architecture maintained. 28% of these also showed evidence ranging from peri-portal fibrosis to cirrhosis (P13, P19, P21, P24, P31, P33). None had any tumors or malignancies. Only three had spleen enlargement measuring from 4 to 6cms below the left costal margin on ultrasound scan. None of these had any other evidence of portal hypertension. Three of the affected (P13, P24 and P33) also had cardiac involvement (ECG and echocardiography) with bi-ventricular hypertrophy. Serum liver enzymes (ALT and AST) were elevated in all. Lipid profile was available in 23 affected and all had raised serum cholesterol and triglyceride levels and a majority had a 2–5 fold increase and three had triglyceride levels over 1000 mg/dl. Fasting hypoglycemia with elevated serum lactate values were documented in 14 (56%) at diagnosis and 10 of them had previous hypoglycemic seizures. Elevated serum CPK (creatine phosphokinase) was documented in 17/21 (80.9%) tested. Thirteen (52%) of the affected had motor delay, generalized hypotonia, Electroneuromyography (ENMG) abnormalities and

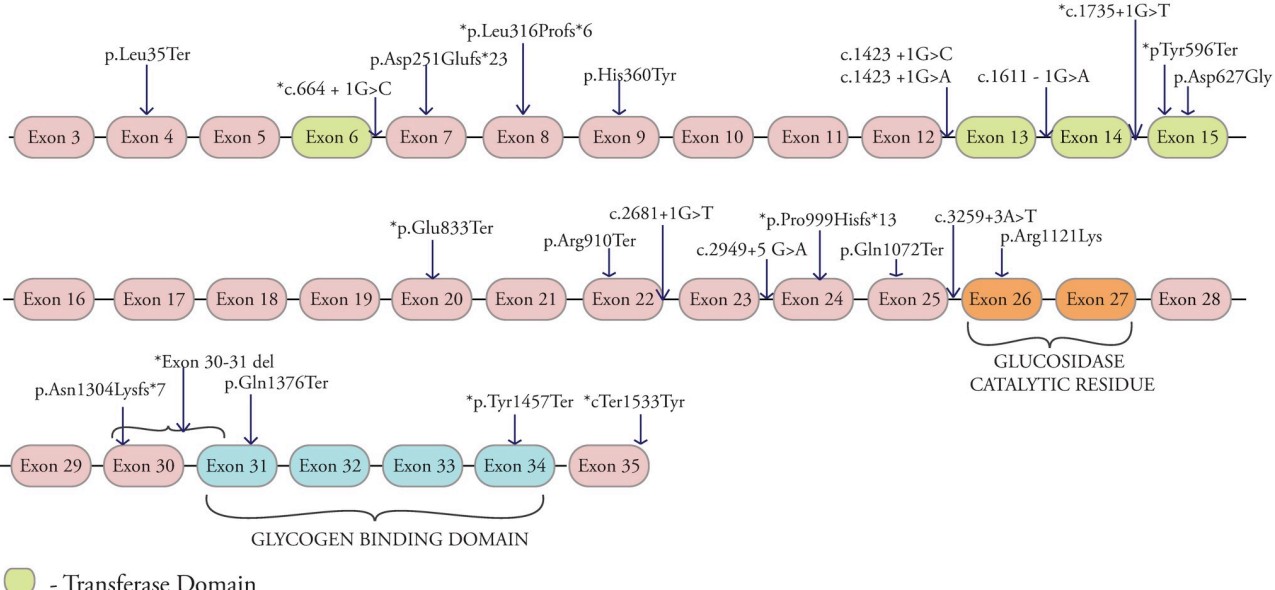

**Fig 2. Distribution of identified pathogenic variants across the *AGL* gene and its domains, \*-novel variations.**

three of these had evidence of cardiac muscle hypertrophy. It was difficult to classify affected individuals into Type IIIa or IIIb, because of young age, absence of exon 3 variations (commonly seen in association with GSDIIIb) and overlapping biochemical findings. However, we believe that 13 (46%) individuals with both hepatic and muscle involvement were likely to have GSDIIIa. All diagnosed with GSDIII (except three not available for regular review), are on dietary intervention with supplementation of uncooked corn starch at frequent intervals and high protein as recommended (GSD III variation list and biochemical parameters available in Table 9 in S1 File).

Nonsense variation c.104T>A [p.Leu35Ter] were found in 2 unrelated individuals (P11 and P12). Nonsense homozygous variations were also noted in individuals P13-P18 (Table 9 in S1 File). A 4bp homozygous deletion was seen in individual P19. P20, P21 and P22 harboured novel homozygous deletions in exon 8, exon 24 and exon 30–31 respectively. All deletions resulted in altered protein sequences. P23 had a novel homozygous insertion C.3903_3904insA [p.Asn1304Lysfs*7] with a frameshift change leading to premature termination of the protein.

Eleven individuals (P24-P33) were noted to have bi-allelic intronic variations of which 3 are novel (P25, P28, P31 and P32). P25 had a novel variation c.664+1G>C in intron 6 and previous study also reported a different base change at the same site which was pathogenic [22]. P28 harboured c.1612-1G>A in the putative transferase catalytic residue which is predicted to alter splicing, leading to exon skipping. P31 and 32 both had a novel homozygous c.2949+5G>A change in exon 23 which causes exon skipping and alteration of the exon enhancer site.

P29 and P30 had a homozygous splice site variation (c.2681+1G>T) in intron 22. Individuals P34, P35 and P36 had two sets of homozygous missense variations each, c.1080C>T [p. His360Tyr] in exon 9 and p.Asp627Gly in exon 15. P37 and P38 had novel homozygous missense variations c.3362G>A [p.Arg1121Lys] in exon 25 c.4599G>C [p.*1533Tyrext*58] in exon 35 respectively. The variation p.Arg1121Lys is in the glucosidase catalytic domain and bioinformatic tools predict it to be damaging (Table 8 in S1 File). The nonstop variation

p.*1533Tyrext*58 replaces the stop codon with tyrosine and the mRNA is read through the 3' region leading to Non-Stop decay [23].

## Glycogen storage disorder type VI

GSDVI results from pathogenic variations in the *PYGL* gene which consists of 20 exons and comprises 813 amino acids (NP_002854.3). Five individuals (P39-P43) harboured a splice site homozygous variation IVS13+1G>C [c.1620+1G>C] which is known to be pathogenic as it generates a transcript lacking all or part of exon 13 [12, 24]. P41 had a compound heterozygous variation c.1620+1G>C and p.Asn24Lys (novel). P44 harboured a novel homozygous duplication c.33Dup [p.Arg12Alafs*99] in exon 1 which results in a frameshift change and premature truncation of the protein 99 amino acids downstream. P45 had a previously reported homozygous missense variation c.2017G>A [p.Glu673Arg] and P46 harboured a homozygous novel missense variation c.2056G>C [p.Gly686Arg] in exon 17.

## Glycogen storage disorder type IXa, b and c

Two individuals P47 and P48 had X-linked variations in *PHKA2*. P47 had a known pathogenic homozygous missense variation c.134G>A [p.Arg45Gln] in exon 2 previously reported in a Chinese study [25] and P48 had a homozygous missense variation c.2870A>G [P.His957Arg] of unknown significance and *in-silico* predictions of which are damaging(scores in Table 8 in S1 File). This individual's history showed hospitalisation thrice for acute gastroenteritis with dehydration, acute prerenal kidney injury, hypoglycaemia, metabolic acidosis and failure to thrive.

P49, P50 and P51 were classified as GSDIXb. Two individuals P49 and P50 harboured large novel deletions (size ~104.30kb.), in exon 2–10 (P49) and exons 2-11(P50). P51 had a novel splice site variation c.1364-2A>G[IVS14-2] in intron 14 affecting the invariant AG acceptor splice site upstream of exon 14 and is predicted to cause exon skipping.

Three unrelated subjects (P52, P53 and P54) had a previously reported pathogenic homozygous missense variation c.317T>A [p.Val106Glu] in exon 4 of *PHKG2* [28]. P55 harboured a novel missense variation, c.229G>A (p.Glu77Lys) which is a conserved codon in the helix region of the protein kinase. P56 had a pathogenic variant c.643G>A[p.Asp215Asn] reported previously [26]. P57 had a homozygous novel deletion c.538Del [p.Pro180Leufs*ter15] in *PHKG2*. Surprisingly, this female now aged 21y has normal growth and menarche and has normal metabolic parameters with dietary intervention from the age of 11y.

## Discussion

This study is the largest from India and the first one using a targeted gene panel to assess GSDs. The majority of affected individuals presented with progressive liver enlargement to their local medical facilities. As genetic testing is not uniformly available in other centres, liver biopsy was performed in the majority (84%) for diagnosis. Children with liver biopsy findings suggestive of GSD were referred to our Centre for genetic testing and further management. In 16% of cases, either because of direct referral or being siblings of previously diagnosed GSD cases, genetic testing was preferred over the more invasive liver biopsy. We identified a total of 49 variations in 57 individuals with GSD. Of these, 27 variations have not been previously reported in the 1000 genome project, dbSNP database and were not present in the 100 control samples screened at our centre. These 27 variations (55%) observed in this study are therefore considered to be novel and were present in highly conserved amino acid positions when analysed across species. Novel missense and splice site variations were predicted deleterious by bioinformatic tools. We initially used Sanger sequencing of the exonic regions in the most

reported GSDs (GSDIa, GSDIb and GSDIII) in individuals who matched these phenotypes. By this method we were able to identify 21variations in 23 individuals. We were later able to utilise a targeted NGS GSD panel covering 23 genes. Using this technique, we identified 28 different variations in 34 individuals across all GSDs reported.

This enabled us to accurately subtype our GSD cohort, perform genotype-phenotype correlations, offer reliable genetic counseling and dietary therapy as appropriate.

Parental consanguinity in our cohort was seen in 42 probands (76%). The rate of consanguinity in the southern Indian population is nearly 20%-30% [27] which may, along with the use of targeted sequencing explain the high number of GSDs identified in our region. The most frequent subtypes in our study based on molecular confirmation were GSDIII- 49%, GSDIX- 19%, GSDVI- 14%, GSDIa-10% and GSDIb- 8%. In a recent Chinese study (49 individuals) the most frequent was GSD1a (45%) and only one variation was found in common with our study (*AGL* c.1735+1G>C) [28]. In an Iranian study (15 individuals), GSD III was the most common subtype reported with parental consanguinity seen in all but 2 families [29], with a single variation found in common with our study (*AGL* p.D251Efs*23). Both these studies also used targeted panel sequencing for diagnosis. A study from Spain (47 individuals) using exome sequencing to screen GSDs found GSDIII with premature termination of the protein being the most common variation [30].

In our study, variations Lys76Met and His119Asp in the functionally active site of *G6PC1* were previously reported with different variations at the same sites [4, 5, 31]. The H119D affects the polarity of this region and is predicted to abolish enzyme activity. All the histidine residues in the protein are placed outside the lumen including H119, which is located at the entry of the lumen membrane. Histidine provides a proton to release the glucose molecules and is also suggested to be the phosphate acceptor site during catalysis [4]. Hence, when replaced by a negatively charged amino acid it is likely to cause altered G6Pase activity. Segregation of the variant in both parents and supportive bioinformatic predictions (Table 8 in S1 File) indicate this variation as disease causing. This patient is now 15 years of age and has normal growth and biochemical parameters on dietary therapy. The novel variation c.227A>T [p. Lys76Met] observed in exon 1 is considered to be 'likely pathogenic' as Lys76 is the active site in the PAP2 superfamily domain of the G6Pase protein. There is another variation reported in HGMD at the same position [31]. It is interesting to note that mutational hotspots p.Arg83Cys and p.Gln347X in Caucasians and the p.Arg83His seen in the Chinese population where not observed in our cohort or other Indian studies [32].

A novel missense variation, c.139G>C [p.Asp47His] in exon 2 of *SLC37A4* leads to conversion of aspartic acid to histidine which is likely to alter the polarity of this region. This amino acid occupies a crucial position, which when converted to histidine could affect the structure and function of the protein.

The largest subset of GSDs in this study was GSDIII (*AGL*). In this sub-type, 3 of the 7 nonsense variations were novel [Tyr596X, Gln833X and Tyr1457X]. A few of the nonsense variations seen in our study were also previously reported in different populations, Leu35X in the Spanish [30], Arg910X in the Italian [33], and Gln1376X in the Turkish populations [34], respectively. The homozygous splice site variation (c.2681+1G>T) observed in intron 22 is a reported pathogenic variation and seen in nearly 28% of GSDIII identified individuals in the Italian and Mediterranean studies [10, 34]. The homozygous insertion of nucleotide A in exon 29 (c.3903_3904) seen in one individual has previously been reported in the Ashkenazi Jewish population [35]. The intronic variation (c.1735+1G>T in intron 14) affects the invariant GT donor splice site of exon 14 causing skipping of exon 15 and has been reported previously in individuals of Chinese, Japanese and Korean ethnicity [22].

A missense variation Asp627Gly was often seen along with another pathogenic variant in our study (Table 9 in S1 File) among which 4 individuals were homozygous and 3 were heterozygous for this variation. One hundred control samples were screened for this variation, and we found 1% to be homozygous and 6% heterozygous, indicating that the Asp627 change is unlikely to be disease-causing. A previous study reported this variation Asp627Gly to be disease causing [9], and an Italian study reported compound heterozygous changes where one of the variations was Asp627Tyr and presented with a mild phenotype [36]. In this GSDIII series, 44% of variations caused premature termination of the protein (nonsense, deletions and insertions), 36% were splice site changes and 16% were missense variations.

The splice site homozygous variation IVS13+1G>C [c.1620+1G>C] in *PYGL* (GSDVI) which was observed in 5 individuals in our study was classified as a founder effect variation as it was observed in 3% of Mennonites who are carriers for this intronic variation and 0.01% who were said to be affected [3]. This variation was also seen in 1 individual from another Indian study [37]. In our study, individuals with the homozygous IVS13+1G>C variation had mild symptoms and responded well to uncooked starch supplementation every 4–6 hourly which is similar to the other reports [3, 38]. The oldest, now aged 16y has growth parameters between the 10-25th centile for age, liver span 8cm, normal liver function tests and is managing well in mainstream school (grade 10).

Two of the 11 GSDIX cases in our study were caused by pathogenic variations in *PHKA2* whereas in other studies, *PHKA2* accounted for up to 75% [3, 13]. GSDIXb (*PHKB*) has not been previously reported in any Indian study. We report 3 GSDIXb cases in our series, 2 of which had large deletions (smaller deletions in *PHKB* gene have been previously reported [27]) and 1 with a splice site variation all of which are novel. Six individuals (54%) had variations in *PHKG2* gene causing GSDIXc was the most common of the type IX GSDs. One variation, V106E seen in three individuals. Seven out of 11 individuals diagnosed with type IX GSD were male and this concurs with previous publications. GSDIX individuals usually present with milder symptoms and are not seen for regular reviews as frequently as other subtypes.

As GSDs usually present with very few distinguishing clinical and biochemical features, it is often challenging in deriving genotype-phenotype correlations and classifying them into clinical subtypes without molecular genetics. Our study and most other Indian studies confirm that variations in *AGL*, *G6PC1* and *SLC37A4* are spread throughout the gene, with few recurring variations. The exception was the splice site variation c.1620+1G>C seen in *PYGL* gene in five unrelated individuals which may indicate a hotspot or founder effect variation in the southern Indian population. In view of the above and because more than around 26 different genes are involved in the etiology of GSDs, we recommend that targeted gene panels are efficient in diagnosing GSDs.

## Supporting information

**S1 File. Figures and tables.**
(DOCX)

## Author Contributions

**Conceptualization:** Swathi Shetty.

**Data curation:** Tejashwini Vittal Kumar, Ambika K. Srikanth, Swathi Anikar.

**Formal analysis:** Tejashwini Vittal Kumar, Ambika K. Srikanth, Swathi Anikar.

**Investigation:** Meenakshi Bhat, Sanjeeva Ghanti Narayanachar, Vinu Narayan.

**Methodology:** Tejashwini Vittal Kumar, Ambika K. Srikanth, Swathi Anikar, Swathi Shetty.

**Project administration:** Swathi Shetty.

**Resources:** Sanjeeva Ghanti Narayanachar, Vinu Narayan.

**Software:** Tejashwini Vittal Kumar.

**Supervision:** Swathi Shetty.

**Visualization:** Swathi Shetty.

**Writing – original draft:** Tejashwini Vittal Kumar, Swathi Shetty.

**Writing – review & editing:** Meenakshi Bhat, Swathi Shetty.

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
