## [Decision Letter · Decision Letter 0]

11 Apr 2022

PONE-D-22-06833Molecular and clinical profiling in a large cohort of Asian Indians with glycogen storage disordersPLOS ONE

Dear Dr. Shetty,

Thank you for submitting your manuscript to PLOS ONE. After careful consideration, we feel that it has merit but does not fully meet PLOS ONE’s publication criteria as it currently stands. Therefore, we invite you to submit a revised version of the manuscript that addresses the points raised during the review process. Besides the important reviewers' comments shown below, authors should also address the following points:In the introduction, please clarify the knowledge gap and study objectives. Please, follow reporting guidelines, such as STREGA) (Little J, Higgins JP, Ioannidis JP, Moher D, Gagnon F, von Elm E, Khoury MJ, Cohen B, Davey-Smith G, Grimshaw J, Scheet P, Gwinn M, Williamson RE, Zou GY, Hutchings K, Johnson CY, Tait V, Wiens M, Golding J, van Duijn C, McLaughlin J, Paterson A, Wells G, Fortier I, Freedman M, Zecevic M, King R, Infante-Rivard C, Stewart A, Birkett N; STrengthening the REporting of Genetic Association Studies. STrengthening the REporting of Genetic Association Studies (STREGA): an extension of the STROBE statement. PLoS Med. 2009 Feb 3;6(2):e22. doi: 10.1371/journal.pmed.1000022.)In methodology, please, clarify study design, setting, and period of recruitment/data collection.Provide diagnostic criteria used for GSD before genetic testing.Please, describe all genetic variants according to HGVS criteria and follow American College of Medical Genetics and Genomics Guidelines for variants classification.Authors should summarize the currently long results section (providing main findings), avoiding repetition of data already present tables. Moreover, authors should not provide interpretations or discussions of findings in the results section (this should be done in the discussion).Table 3 is very condensed; it is better to provide it as a supplementary and provide a summary of the most important findings in the main text.Authors have to discuss study limitations and generalizability.

We look forward to receiving your revised manuscript.

Kind regards,

Elsayed Abdelkreem, MD, PhD

Academic Editor

PLOS ONE

Journal Requirements:

Reviewers' comments:

Reviewer's Responses to Questions

**Comments to the Author**

1. Is the manuscript technically sound, and do the data support the conclusions?

Reviewer #1: Yes

Reviewer #2: Yes

Reviewer #3: Yes

Reviewer #4: Yes

Reviewer #5: Yes

2. Has the statistical analysis been performed appropriately and rigorously? 

Reviewer #1: N/A

Reviewer #2: Yes

Reviewer #3: N/A

Reviewer #4: N/A

Reviewer #5: Yes

3. Have the authors made all data underlying the findings in their manuscript fully available?

Reviewer #1: Yes

Reviewer #2: Yes

Reviewer #3: Yes

Reviewer #4: Yes

Reviewer #5: Yes

4. Is the manuscript presented in an intelligible fashion and written in standard English?

Reviewer #1: Yes

Reviewer #2: Yes

Reviewer #3: Yes

Reviewer #4: Yes

Reviewer #5: Yes

5. Review Comments to the Author

Reviewer #1: Specific comments to authors:

Kumar and co-others in this research article report on ‘‘Molecular and clinical profiling in a large cohort of Asian Indians with glycogen storage disorders’’. Congratulations on performing this outstanding research. The manuscript is properly written, and of clinical interest although the authors need to address some points as follows:

Major considerations:

- Please place the methods section before the results.

- In table 1, compare the phenotypes of different GSDs types by percentage, mean, median and P-value to be more representative.

- Please add to table 1 the different associated affected systems (muscular, renal, GIT).

- Mention the growth parameters of the patients in different genotypes and how much they are affected according to growth percentiles.

- Mention the age of death of the deceased patients.

- You mentioned the dietary regimens for patients with GSD III. What about type I, VI and IX in your cohort?

- In table 3: the heading of the 2nd column is incorrect. Did any of your patients with GSD III have splenomegaly, especially the ones with liver fibrosis? The absolute figures of weight and height are not representative. Please add the percentiles.

- Did any of the patients with GSD III have liver steatosis in liver biopsy, which is a common finding in this type?

- You mentioned that the surviving patients are doing fine!! This is a vague sentence. Please mention is short the mean duration of follow up of your patients and their prognosis regarding hepatomegaly, growth affection, and metabolic control.

- Focus in the results on your findings and not too much on the theoretical background.

- Explain why did you perform a liver biopsy for your patients which is an invasive procedure with its hazards and not conclusive while they have a genetic testing? Or the genetic testing was available later on?

- Did any of your patients have a prenatal diagnosis as you mentioned in the methods? What is its value unless a therapeutic abortion was planned?

- I wonder if you excluded any patients who had a similar clinical and phenotypic picture as GSD and turned out not to be GSD (e.g. fatty acid oxidation defects, fructose 1,6 biphosphatase deficiency, some types of CDGs). This will add to the value of your work in offering genetic testing in these suspected cases and not to depend only on the clinical and biochemical profile.

Minor considerations:

- The manuscript needs revision for minor editing and grammar mistakes.

- Please stick to either the English of American style in writing the whole manuscript.

Thanks

Reviewer #2: It was very interesting to read the manuscript titled: "Molecular and clinical profiling in a large cohort of Asian Indians with glycogen storage disorders". The manuscript is very well, describing the results of genetic testing in an Indian cohort of patients with GSD over a 15-year period. The results are very well presented.

Reviewer #3: The authors presented a good study.

1. Please consult ACMG recommendations to classify each newly identified variant (Ellard 2020). It is highly important to have enough evidence in order to classify variant as “pathogenic”. Please use accordingly pathogenic, likely pathogenic and VUS (classes 1, 2 and 3) for each new variant.

2. The manuscript would benefit from shortening the Introduction section.

3. Page 4, line 96, Correct typo: SLC37A4, not SLC37A41

Reviewer #4: The authors reported 57 patients with glycogen storage disease who underwent genetic analyses in India. This study provides important evidence for ethnic founder effects in the Indian population. This manuscript includes the interest of the readers of the journal. However, this paper does not follow the manner of publishing paper, as below.

There are too much textbook descriptions in the 'Introduction' part. Please omit the general description of glycogen storage diseases and provide the introduction on the founder effects in India and other regions.

There is no distinction between results and discussion. The results part should not include the authors' interpretations but only the facts. Please compare with previous reports and describe the authors' interpretations in the discussion section.

Criteria for determining novel mutations as pathogenic is ambiguous and not uniform. Online bioinformatic tools are helpful but only secondary evidence, so the American College of Medical Genetics and Genomics Guidelines should be used to decide. If possible, please add the values of each enzyme activity.

Please describe the details of the supportive bioinformatic predictions (Page 9, Lines 179-180), the bioinformatic tools predicting (Page 20, Line 270), and in-silico predictions (Page 21, Line 295).

Page 9, Lines 186-188

The notation 'likely pathogenic' is the same as the description of P02. Please clarify whether this is the authors' opinion or a quote from the NCBI database.

Page 9, Lines 190-191

This description only tells us that they had cirrhosis, not whether they had glycogen storage disease.

Page 11, Table III

Normal ranges for height, weight, and liver size vary with age. Please describe whether these values are normal.

Reviewer #5: Congratulations for detailed and nice results

The authors have evaluation the GSD patients with sanger sequencing and they have subtyped the cases in detail. This is a well organized and well written paper.

6. PLOS authors have the option to publish the peer review history of their article (what does this mean?). If published, this will include your full peer review and any attached files.

Reviewer #1: **Yes: **Engy Mogahed

Reviewer #2: **Yes: **Hanaa El-Karaksy

Reviewer #3: **Yes: **Maja Stojiljkovic

Reviewer #4: No

Reviewer #5: **Yes: **Bita Geramizadeh

---

## [Author Response · Author response to Decision Letter 0]

20 May 2022

Please find below our responses/changes made to the editors and reviewers comments 

- Besides the important reviewers' comments shown below, authors should also address the following points:

• In the introduction, please clarify the knowledge gap and study objectives.

- Response: We have added a few lines to address this. (lines 115-122)

- Please, follow reporting guidelines, such as STREGA) (Little J, Higgins JP, Ioannidis JP, Moher D, Gagnon F, von Elm E, Khoury MJ, Cohen B, Davey-Smith G, Grimshaw J, Scheet P, Gwinn M, Williamson RE, Zou GY, Hutchings K, Johnson CY, Tait V, Wiens M, Golding J, van Duijn C, McLaughlin J, Paterson A, Wells G, Fortier I, Freedman M, Zecevic M, King R, Infante-Rivard C, Stewart A, Birkett N; STrengthening the REporting of Genetic Association Studies. STrengthening the REporting of Genetic Association Studies (STREGA): an extension of the STROBE statement. PLoS Med. 2009 Feb 3;6(2):e22. doi: 10.1371/journal.pmed.1000022.)

- Response: We have complied with STREGA guidelines as suggested.

-In methodology, please, clarify study design, setting, and period of recruitment/data collection.

- Response :This has been included. (Line 134-135)

- 

- Provide diagnostic criteria used for GSD before genetic testing.

- Response: Line 134-142 has the necessary information

-Please, describe all genetic variants according to HGVS criteria and follow American College of Medical Genetics and Genomics Guidelines for variants classification.

- Response: This has been done as suggested. (ref: Table I and supplementary file)

-Authors should summarize the currently long results section (providing main findings), avoiding repetition of data already present tables. Moreover, authors should not provide interpretations or discussions of findings in the results section (this should be done in the discussion).

- Response: This has been changed to the extent possible.

-Table 3 is very condensed; it is better to provide it as a supplementary and provide a summary of the most important findings in the main text.

-Response: ,As suggested we agree to put table III as supplementary data. The summary of the most important findings from this table are already incorporated in the main text under result section ‘GSDIII’ subheading.

- Authors have to discuss study limitations and generalizability.

- Response: We cannot think of any limitations other than cost and diagnosing based on variable clinical phenotype which we have encountered and mentioned in the paper. Similarly with generalizability, the technology employed, difficulty in genotype phenotype correlations have been mentioned.

Review Comments to the Author

Reviewer #1: Specific comments to authors:

Kumar and co-others in this research article report on ‘‘Molecular and clinical profiling in a large cohort of Asian Indians with glycogen storage disorders’’. Congratulations on performing this outstanding research. The manuscript is properly written, and of clinical interest although the authors need to address some points as follows:

Major considerations:

- Please place the methods section before the results.

- Response: This has been done.

- In table 1, compare the phenotypes of different GSDs types by percentage, mean, median and P-value to be more representative.

- Please add to table 1 the different associated affected systems (muscular, renal, GIT).

- Mention the growth parameters of the patients in different genotypes and how much they are affected according to growth percentiles.

- Response: We thank the reviewer for the three suggestions above. Table 1 details the novel variations in GSD subtypes noted in this study without any clinical details included. The authors of this manuscript are also compiling a separate manuscript specifically detailing individual clinical phenotypes, systems involved and severity, percentages and statistical frequencies in all GSD sub-types studied so far. They will also discuss the long-term clinical profiling and outcomes in affected individuals on dietary and other interventions, highlighting improvements gained from intervention over time. We wish to submit that adding a large volume of additional detail on clinical phenotypes, disease course and the ensuing discussion in this manuscript may dilute the clarity of the already large volume of data presented here. It is the intent of the authors to focus on the molecular profiling with emphasis on structure of novel variations in relation to the presenting clinical symptomatology in this write-up.

- Mention the age of death of the deceased patients.

- Response: This has been added in text for both Type 1a (line no 227-228) and 1b (line no 241) in text. 

- You mentioned the dietary regimens for patients with GSD III. What about type I, VI and IX in your cohort?

- Response: The numbers of individuals on dietary therapy with uncooked corn starch is listed in Table II for the various GSD subtypes. A total of 37 individuals in various GSD subtypes are on tailored dietary therapy

- In table 3: the heading of the 2nd column is incorrect. 

- Response: This has been changed. Thank you

-Did any of your patients with GSD III have splenomegaly, especially the ones with liver fibrosis? The absolute figures of weight and height are not representative. Please add the percentiles.

- Response: We have added the numbers of cases with spleen enlargement. The weight and heights have now been represented by their centiles as suggested in Table 10 of supplementary file.

- Did any of the patients with GSD III have liver steatosis in liver biopsy, which is a common finding in this type?

- Response: The majority of patients where liver biopsy was done in GSD III had enlarged distended hepatocytes with pale cytoplasm, staining positive with PAS and diastase resistant indicating the presence of glycogen. The liver biopsy slides were reviewed and none were found to show liver steatosis.

- You mentioned that the surviving patients are doing fine!! This is a vague sentence. Please mention is short the mean duration of follow up of your patients and their prognosis regarding hepatomegaly, growth affection, and metabolic control.

- Response: This has been altered as suggested.

- Focus in the results on your findings and not too much on the theoretical background.

- Response: We have moved some parts of the results into the discussion to address the above. I hope this is now acceptable.

- Explain why did you perform a liver biopsy for your patients which is an invasive procedure with its hazards and not conclusive while they have a genetic testing? Or the genetic testing was available later on?

- Response: The reasons for large numbers of affected individuals with a liver biopsy done prior to genetic testing is now added in the first paragraph of discussion (Line no 341-345).

- Did any of your patients have a prenatal diagnosis as you mentioned in the methods? What is its value unless a therapeutic abortion was planned?

- Response: Yes, a number of families opted for prenatal diagnosis with the intent of medical termination of an affected pregnancy. Details of this has been added under results (Line no 190-195).

- I wonder if you excluded any patients who had a similar clinical and phenotypic picture as GSD and turned out not to be GSD (e.g. fatty acid oxidation defects, fructose 1,6 biphosphatase deficiency, some types of CDGs). This will add to the value of your work in offering genetic testing in these suspected cases and not to depend only on the clinical and biochemical profile.

- Response: The clinical features and ages at presentation in the differential diagnosis to GSD allowed us to include TMS for early biochemical derangement in the majority, which were then followed up by exome sequencing. Two cases of Fructose1,6, biphosphatase deficiency and one each of primary fatty oxidation defect and CDG in early infancy were molecularly confirmed and not included in this study. All these are on regular review at our centre

Minor considerations:

- The manuscript needs revision for minor editing and grammar mistakes.

- Response: We have made changes.

- Please stick to either the English of American style in writing the whole manuscript.

- Response: We have kept to the UK English style.

Reviewer #2: It was very interesting to read the manuscript titled: "Molecular and clinical profiling in a large cohort of Asian Indians with glycogen storage disorders". The manuscript is very well, describing the results of genetic testing in an Indian cohort of patients with GSD over a 15-year period. The results are very well presented.

Reviewer #3: The authors presented a good study.

1. Please consult ACMG recommendations to classify each newly identified variant (Ellard 2020). It is highly important to have enough evidence in order to classify variant as “pathogenic”. Please use accordingly pathogenic, likely pathogenic and VUS (classes 1, 2 and 3) for each new variant.

- Response: We have classified the variants according to the ACMG guidelines.

2. The manuscript would benefit from shortening the Introduction section.

- Response: We have reduced the introduction part as much as possible without making it sound abrupt. I hope this is fine.

3. Page 4, line 96, Correct typo: SLC37A4, not SLC37A41

- Response: Corrected, Thank you.

Reviewer #4: The authors reported 57 patients with glycogen storage disease who underwent genetic analyses in India. This study provides important evidence for ethnic founder effects in the Indian population. This manuscript includes the interest of the readers of the journal. However, this paper does not follow the manner of publishing paper, as below.

-There are too much textbook descriptions in the 'Introduction' part. Please omit the general description of glycogen storage diseases and provide the introduction on the founder effects in India and other regions.

- Response: We have reduced the introduction part as much as possible without detracting from the readability and flow. We have included the one other plausible founder effect variation published and mentioned the other Indian studies published. I hope this is fine.

-There is no distinction between results and discussion. The results part should not include the authors' interpretations but only the facts. Please compare with previous reports and describe the authors' interpretations in the discussion section.

- Response: We have moved parts of our results that include discussions into the discussion section.

-Criteria for determining novel mutations as pathogenic is ambiguous and not uniform. Online bioinformatic tools are helpful but only secondary evidence, so the American College of Medical Genetics and Genomics Guidelines should be used to decide. If possible, please add the values of each enzyme activity.

- Response: This has been done. Measurement of enzyme activity is, unfortunately not available in our centre.

-Please describe the details of the supportive bioinformatic predictions (Page 9, Lines 179-180), the bioinformatic tools predicting (Page 20, Line 270), and in-silico predictions (Page 21, Line 295).

- Response: These are presented in detail in the supplementary files along with scores for each bioinformatic tool used.

Page 9, Lines 186-188

-The notation 'likely pathogenic' is the same as the description of P02. Please clarify whether this is the authors' opinion or a quote from the NCBI database.

- Response: Clarified in the text.

Page 9, Lines 190-191

-This description only tells us that they had cirrhosis, not whether they had glycogen storage disease.

- Response: Thank you for highlighting this, this has been modified with additional details of liver biopsy findings.

Page 11, Table III

-Normal ranges for height, weight, and liver size vary with age. Please describe whether these values are normal.

- Response: The age related centiles for growth parameters and the age based increase in liver spans have been clarified in the table as suggested. I hope this should be satisfactory

Reviewer #5: Congratulations for detailed and nice results

The authors have evaluation the GSD patients with sanger sequencing and they have subtyped the cases in detail. This is a well organized and well written paper.

---

## [Decision Letter · Decision Letter 1]

9 Jun 2022

Molecular and clinical profiling in a large cohort of Asian Indians with glycogen storage disorders

PONE-D-22-06833R1

Dear Dr. Shetty,

We’re pleased to inform you that your manuscript has been judged scientifically suitable for publication and will be formally accepted for publication once it meets all outstanding technical requirements.

Kind regards,

Elsayed Abdelkreem, MD, PhD

Academic Editor

PLOS ONE

Additional Editor Comments (optional):

Reviewers' comments:

Reviewer's Responses to Questions

**Comments to the Author**

1. If the authors have adequately addressed your comments raised in a previous round of review and you feel that this manuscript is now acceptable for publication, you may indicate that here to bypass the “Comments to the Author” section, enter your conflict of interest statement in the “Confidential to Editor” section, and submit your "Accept" recommendation.

Reviewer #1: All comments have been addressed

Reviewer #2: All comments have been addressed

Reviewer #3: All comments have been addressed

Reviewer #4: All comments have been addressed

2. Is the manuscript technically sound, and do the data support the conclusions?

Reviewer #1: Yes

Reviewer #2: Yes

Reviewer #3: Yes

Reviewer #4: Yes

3. Has the statistical analysis been performed appropriately and rigorously? 

Reviewer #1: Yes

Reviewer #2: N/A

Reviewer #3: N/A

Reviewer #4: N/A

4. Have the authors made all data underlying the findings in their manuscript fully available?

Reviewer #1: Yes

Reviewer #2: Yes

Reviewer #3: Yes

Reviewer #4: Yes

5. Is the manuscript presented in an intelligible fashion and written in standard English?

Reviewer #1: Yes

Reviewer #2: Yes

Reviewer #3: Yes

Reviewer #4: Yes

6. Review Comments to the Author

Reviewer #1: All points have been addressed and the manuscript in my opinion is suitable for publication.

Thanks

Reviewer #2: Well written manuscript, interesting work. There are no additional comments. All comments have been addressed

Reviewer #3: (No Response)

Reviewer #4: The authors have adequately addressed my comments. This is a useful paper for world-wide researchers besides indeian clinicians.

7. PLOS authors have the option to publish the peer review history of their article (what does this mean?). If published, this will include your full peer review and any attached files.

Reviewer #1: **Yes: **Engy Adel Mogahed

Reviewer #2: No

Reviewer #3: **Yes: **Maja Stojiljkovic, IMGGE, full research professor, University of Belgrade

Reviewer #4: **Yes: **Mitsuru Kubota

---

## [Editor Report · Acceptance letter]

17 Jun 2022

PONE-D-22-06833R1 

Molecular and clinical profiling in a large cohort of Asian Indians with glycogen storage disorders 

Dear Dr. Shetty:

I'm pleased to inform you that your manuscript has been deemed suitable for publication in PLOS ONE. Congratulations! Your manuscript is now with our production department. 

Kind regards, 

on behalf of

Dr. Elsayed Abdelkreem 

Academic Editor

PLOS ONE